# Old and Promising Markers Related to Autophagy in Traumatic Brain Injury

**DOI:** 10.3390/ijms24010072

**Published:** 2022-12-21

**Authors:** Tommaso Livieri, Calogero Cuttaia, Raffaella Vetrini, Monica Concato, Michela Peruch, Margherita Neri, Davide Radaelli, Stefano D’Errico

**Affiliations:** 1Department of Medical Surgical and Health Sciences, University of Trieste, 34149 Trieste, Italy; 2Department of Medical Sciences, Section of Public Health Medicine, University of Ferrara, 44121 Ferrara, Italy

**Keywords:** traumatic brain injury, autophagy, mitophagy, autopsy, immunohistochemistry

## Abstract

Traumatic brain injury (TBI) is one of the first causes of death and disability in the world. Because of the lack of macroscopical or histologic evidence of the damage, the forensic diagnosis of TBI could be particularly difficult. Considering that the activation of autophagy in the brain after a TBI is well documented in literature, the aim of this review is to find all autophagy immunohistological protein markers that are modified after TBI to propose a method to diagnose this eventuality in the brain of trauma victims. A systematic literature review on PubMed following PRISMA 2020 guidelines has enabled the identification of 241 articles. In all, 21 of these were enrolled to identify 24 markers that could be divided into two groups. The first consisted of well-known markers that could be considered for a first diagnosis of TBI. The second consisted of new markers recently proposed in the literature that could be used in combination with the markers of the first group to define the elapsed time between trauma and death. However, the use of these markers has to be validated in the future in human tissue by further studies, and the influence of other diseases affecting the victims before death should be explored.

## 1. Introduction

The common definition of traumatic brain injury (TBI) is cerebral damage caused by a traumatic event and it is one of the first causes of death and disability in the world. Motor vehicle traffic accidents or sport traumas in young people and accidental falls in older people are the main cause of TBI. It is possible to distinguish a primary injury, caused by an external force acting directly on the head, and a secondary injury, caused by lesion evolution through cellular damage such as ischemia and edema. As symptomatology is largely clinic, there is often little or absent macroscopical or histologic evidence of the damage in the case of the victim’s death [1,2] so it is a particularly difficult challenge to make a diagnosis of TBI in the forensic field [3,4,5]. One of the main purposes of the research within TBI is to understand the molecular mechanism behind the TBI process, not only for clinical and therapeutical purposes, but also in the forensic field to accomplish a more precise diagnosis. In particular, while the involvement of apoptosis and necroptosis in both phases of TBI is well known and consolidated, the study of the TBI molecular process proves that autophagy is also active after an encephalic trauma as a mechanism of defense from neuronal damage [6,7,8,9,10,11].

Therefore, considering the proven involvement of autophagy in TBI, the aim of the study is to find all autophagy immunohistological protein markers that were modified after TBI in literature, for the purpose of proposing a method to diagnose this eventuality in the brain of trauma victims. The possibility to determine the timing of death and lesion is also considered.

## 2. Results

The literature review has enabled the identification of two groups of markers: the first was made up of well-known markers (studied in five or more studies), whose correlation with autophagy is consolidated and used today to test new drugs against negative outcomes of TBI; the other was made up of new markers recently proposed in the literature with few dedicated studies (involved in fewer than five studies).

Articles included in the review are listed in Table 1.

In the first group, we find Beclin-1 and lipidated microtubule-associated protein light-chain 3 (LC3-II). In the second group we find Connexine family, glycogen synthase kinase-31β (Gsk-3β), p62, lysosomal-associated membrane proteins (LAMP) family, cathepsin D (CSTD), Wnt1 inducible signaling pathway protein 1 (WISP1), β-catenin, 3-mercaptopyruvate sulfurtransferase (3-MST), transactivation response deoxyribonucleic acid protein 43 (TDP-43), autophagy related 7 (Atg7 interleukin 33 (IL-33), interleukin 1 receptor-like 1 (IL1RL1), senescence-associated-β-galactosidase (SA-β-gal), Cyclin-D, proliferating cell nuclear antigen (PCNA), p16, p21, Chromodomain-helicase-DNA-binding protein 8 (Chd8) and Sec22b. Below they will be treated one by one.

### 2.1. Beclin 1

Beclin-1 is part of the activation complex of macroautophagy and is the human equivalent of the protein Atg6 of yeast. Its action is inhibited by bcl-2. It was the first marker of autophagy that had been searched in post-TBI brains to prove the correlation between TBI and autophagy [6,12]. The first studies demonstrate an increase of the protein 4 h after the trauma [6,10,12] that last for 3 weeks [6,12] or 32 days [10] in the perilesional cortex, with a peak on the eighth day [10]. However, earlier studies found weaker elevation of the protein and its inhibitor, bcl-2, and demonstrate a stronger correlation between TBI timing and Beclin-1/bcl-2 ratio [11,13]. Significantly, Beclin-1 is the only marker researched also in the human brain with LC3-II, but it has brought poor results: in Clark et al. [7] the protein was found elevated in only 1 of the 5 patients. Moreover, the patients that demonstrate a rise of Beclin-1 were not the same that demonstrate a rise of LC3-II. It is interesting that Sebastiani et al. [14] do not find a significant variation of Beclin-1 and its mRNA using respectively western blot (WB) and real time-PCR techniques in TBI mice model in the first seven days after the trauma.

### 2.2. Lipidated Microtubule-Associated Protein Light-Chain 3 (LC3-II)

Lipidated microtubule-associated protein light-chain 3 (LC3-II) is a protein involved in the conjugation phase of macroautophagy and is the human equivalent of the protein Atg8 of the yeast. It is one of the first markers of autophagy studied in the context of TBI and is probably the better known. The increment of the protein starts in the first four hours after the trauma, although the rise is not always significant [8,15]. However, all the studies agree in demonstrating a long elevation of the protein that was found to arise also at 15 days [9] or 32 days [10]. Similarly, the peak of increment of LC3-II is demonstrated to be around 24 h after the trauma [8,13,15,16]. The IHC also consent to determining that the protein is mostly associated with the membrane and the living brain cells [9,13] but also in non-vital neurons [16]. The positivity was found in neurons until the third day and after then in glial cells with a peak on the third day [10,13]. The protein was found particularly in the perilesional and ipsilateral cortex [10,16,17] but also in the ipsilateral hippocampus in the contralateral cortex [11,13,15]. Significantly, LC3-II is the only marker researched also in the human brain with Beclin-1, but brought poor results: in Clark et al. [7] the protein was found increased in 4 of the 5 patients but also in 3 of the 5 controls. Moreover, the patients that demonstrate a rise of LC3-II were not the same that demonstrate a rise of Belcin-1.

**Table 1 ijms-24-00072-t001:** Articles included in the review.

Articles	Markers	Methods	Results
Diskin et al., 2005 [6]	Beclin-1	WB, IHC, and IF in the brain of TBI mice models	Increased in the perilesional cortex and hippocampus. Increased between 4 h and 3 weeks after trauma with a peak at 1 week after trauma in neurons. Increased between 3 days and 3 weeks in astrocytes
Erlich et al., 2006 [12]	Beclin-1	WB, IHC, and IF in the brain of TBI mice models	Increased in the perilesional cortex and hippocampus. Increased between 4 h and 3 weeks after trauma with a peak at 1 week after trauma in neurons. Increased between 3 days and 3 weeks in astrocytes
Clark et al., 2007 [7]	Beclin-1	WB and IF in the brain of humans subjected to decompressive craniotomy after TBI	Increased in 1 of the 5 patients without correlation with LC3-II
	LC3-II	WB and IF in the brain of humans subjected to decompressive craniotomy after TBI	Increased in 4 of the 5 patents and 3 of 5 controls without correlation with Belcin-1
Lai et al., 2008 [8]	LC3-II	WB in the brain of TBI mice models	Increased between 2 and 48 h after the trauma whit a peak at 24 h after the trauma
Liu et al., 2008 [9]	LC3-II	WB and IF in the brain of TBI rat models	Increased non significantly 4 h after trauma and significantly between 1 and 15 days after trauma in living neurons
Zhang et al., 2008 [10]	LC3-II	IHC and IF in the brain of TBI mice models	Increased between 1 h and 32 days after trauma with a peak at 8 h after trauma in the perilesional cortex. Specially in neurons until the third day, then in glial cells
	Beclin-1	WB, IHC, and IF in the brain of TBI mice models	Increased Between 1 h and 32 days after trauma with a peak at 8 days after trauma in perilesional cortex and hippocampus especially in neurons until the third day, then in glial cells
Sadasivan et al., 2008 [11]	LC3-II	WB in the brain of TBI rat models	Increased between 2 h and 2 days in ipsilateral and contralateral brain
	Beclin-1	WB in the brain of TBI rat models	Increased non significantly after 2 and 6 h but the rate of beclin-1/bcl2 increase significantly at 1 and 2 days
Sun et al., 2014 [15]	LC3-II	WB and IF in the brain of TBI rat models	Increased non significantly 3 h after the trauma and significantly 6, 24, and 48 h after trauma with a peak at 24 h after the trauma predominantly in living neurons of the hippocampus
	p-CX43	WB and IF in the brain of TBI rat models	Increased significantly between 3 and 24 h after the trauma with a peak at 6 h after the trauma in the astrocyte of the hippocampus
Lin et al., 2014 [18]	LC3-II	WB in the brain of TBI rat models	Increased between 30 min and 24 h after the trauma
	GSK-3β	WB in the brain of TBI rat models	Active form increased and inactive form decreased between 30 min and 24 h after the trauma
Sarkar et al., 2014 [13]	LC3-II	WB and IF in the brain of TBI mice models	Increased between 1 h and 1 week after trauma with a peak between 1 and 3 days after traumas, at first in neurons and then in glial cells with a peak on third day. In the ipsilateral and contralateral cortex and hippocampus
	Beclin-1	WB in the brain of TBI mice models	No significant variation
	p62	WB and IF in the brain of TBI mice models	Increased between 1 h and 7 days after the trauma in the ipsilateral cortex and hippocampus
	LAMP family	WB in the brain of TBI mice models	LAMP1 and LAMP2 increase between 3 and 7 days after trauma in lysosomal and cytosolic fractions
	CTSD	WB and IF in the brain of TBI mice models	Decreased 1 and 24 h after trauma and increased 3 and 7 days after trauma in the cortex and hippocampus
Park et al., 2015 [19]	LAMP family	WB and IF in the brain of TBI rat models	LAMP2A increases between 1 and 15 days in neurons and glial cells after trauma especially in CA3 and Cx areas ipsilaterally
	HSP70	WB in the brain of TBI rat models	Increased between 1 and 15 days after the trauma with a peak at 3 days after the trauma
Ye et al., 2017 [17]	WISP1	WB and IF in the brain of TBI rat models	Decreased 1 day after trauma, reach the minimum at 3 days after trauma, and recovered at 5 and 7 days in neurons, in particular in the ipsilateral brain
	LC3-II	WB and IF in the brain of TBI rat models	Increased 3 days after trauma in ipsilateral brain
	β-catenin	WB and IF the in brain of TBI rat models	Decreased between 1 and 7 days after trauma with a negative peak at 3 days after trauma and a partial restoration at 5 days after trauma
Zhang et al., 2017 [16]	3-MST	WB, IHC, and IF in the brain of TBI mice models	Increased between 6 h and 3 days after trauma with a peak at 1 day after trauma in vital neurons of the ipsilateral cortex
	LC3-II	WB and IF in the brain of TBI mice models	Increased between 12 h and 2 days after trauma with a peak at 24 h in vital and non-vital neurons in the ipsilateral cortex
Che et al., 2017 [20]	Cx40	WB and IF in the brain of TBI rat models	Increased between 6 h and 6 days after trauma with a peak at 1 day after trauma in neurons of the perilesional cortex, at first in intercellular vesicles, and then diffused in the cell
Sebastiani et al., 2017 [14]	p62	WB in the brain of TBI mice models	Decreased between 1 and 5 days after trauma
Saykally et al., 2018 [21]	TDP-43	Radioimmunoprecipitation in the brain of TBI mice models	Increased 12 days after 1 50 g trauma. Increased 3 days after 5 30 g traumas and still increased, not significantly after 30 days in the ipsilateral cortex. Decreased in the ipsilateral hippocampus between 30 and 60 days after 5 30 g traumas
	LC3-II	Radioimmunoprecipitation in the brain of TBI mice models	Increased 3 days after trauma
	Atg7	Radioimmunoprecipitation in the brain of TBI mice models	Decreased 3 days after trauma
	LAMP family	Radioimmunoprecipitation in the brain of TBI mice models	LAMP2A decreases 60 days after trauma in the ipsilateral hippocampus
Wiesner et al., 2018 [22]	TDP-43	IF in the brain of TBI and stab mice models	Increased between 3 and 7 days after the trauma with a peak at 3 days after the trauma
Gao et al., 2018 [23]	IL-33	WB and IF in the brain of TBI mice models	Increased 1 day after trauma in astrocyte nucleus and oligodendrocyte cytoplasm
	ST2L	WB and IF in the brain of TBI mice models	Decreased 1 day after trauma
Tominaga et al., 2019 [24]	SA-β-gal	IHC and IF in the brain of TBI mice models	Increased between 1 and 14 days after trauma with a peak at 7 days after trauma in the ipsilateral cortex
	Ciclin-D1	IHC and IF in the brain of TBI mice models	Increased between 1 and 14 days after trauma with a peak between 4 and 7 days in neurons, astrocytes, and microglia in the ipsilateral cortex
	PCNA	IHC and IF in the brain of TBI mice models	Increased between 1 and 14 days after trauma with a peak between 4 and 7 days in neurons, astrocytes, and microglia in the ipsilateral cortex
	p16	IHC and IF in the brain of TBI mice models	Increased between 4 and 14 days after trauma in astrocyte of ipsilateral brain
	p21	IHC and IF in the brain of TBI mice models	Increased between 4 and 14 days after trauma in neurons and microglia of the ipsilateral brain
Chen et al., 2020 [25]	Chd8	WB and IF in the brain of TBI mice models	In the perilesional cortex it decreases 3 h after trauma, partially restores at 12 h, and decreases again until the seventh day. In prefrontal cortex increases between 12 and 24 h after trauma with a peak at 12 h. In the hippocampus, it increases between 6 and 24 h with a peak at 12 h. It is localized in neurons
	β-catenin	WB and IF in the brain of TBI mice models	In the prefrontal cortex, it increases between 6 h and 7 days after trauma with a peak at 12 h. In perilesional cortex and hippocampus it increases between 12 and 24 h with a peak at 12 h after trauma
Li et al., 2021 [26]	Sec22b	WB and IF in the brain of TBI rat models	Decreased between 12 h and 7 days after trauma with a negative peak at 24 h after trauma in the perilesional cortex

### 2.3. Connexins Family

Connexins are a family of proteins that form gap junctions allowing the passage of small molecules between cells. Some of them, like Connexin40 (Cx40), Connexin43 (Cx43), and Connexin50 (Cx50) have an important function at neuronal and astrocytic levels and are involved in autophagy, since they allow the propagation of death cell signaling [27,28]. Using WB and immunofluorescence (IF) in the brain of TBI rats models, Sun et al. [15] demonstrate that the phosphorylated form of Connexin43, p-CX43 increases significantly between 3 and 24 h after the traumatic event, with a peak in astrocyte of the hippocampus at 6 h after the traumatic event. Moreover, similar methods in Che et al. [20] show an increase of Cx40 in the neurons between 6 h and 6 days after the trauma, with a peak at 24 h. Moreover, IF demonstrates that the protein is located in the neurons of the perilesional cortex, initially in intracellular vesicles and then diffused in the cell.

### 2.4. Glycogen Synthase Kinase-31β (GSK-3β)

GSK-3β is a serine/threonine kinase involved in cell survival, apoptosis, and autophagy pathways. It has an active form, phosphorylated in tyr216 and dephosphorylated in Ser9, and an inactive form, dephosphorylated at tyr216 and phosphorylated at Ser9 [29]. Lin et al. [18] found that the active form increases between 30 min and 24 h after the trauma using WB in the brain of TBI rat models along with a decrease of the inactivated form.

### 2.5. p62 or Seqeuestrsomoe-1 (SQSTM1)

p62 is a protein that targets ubiquitinated proteins and allows their phagocytosis during the macroautophagy [30,31]. Using WB and IF in the brain of TBI mice models, Sarkar et al. [13] found an increase of the protein between 1 h and 7 days after trauma in the ipsilateral cortex and hippocampus without significant alteration of mRNA in real time-PCR. Besides, Sebastiani et al. [14] found a deficiency of the protein between 1 and 5 days after the trauma using a WB. This study also demonstrates a decrease of p62 mRNA between 3 and 5 days after trauma using real time-PCR.

### 2.6. Lysosomal-Associated Membrane Proteins (LAMPs) and Heat Shock Protein 70 (HSP70)

LAMPs are integral lysosome membrane proteins. It is demonstrated that LAMP2A is involved in Chaperone Mediated Autophagy (CMA) process since they bind misfolded protein that are bound to heat shock protein 79 (HSC79) and allow their phagocytosis [32]. LAMP1 and LAMP2 were found increased at the WB on lysosomal and cytosolic fractions between 3 and 7 days after trauma in Sarkar et al. [13] and Park and al. [19] describe an elevation LAMP2A with HSP70 both in neurons and glial cells of CA3 and Cx hippocampus areas of the brain bilaterally between 1 and 15 days in a TBI in mice models studied with electrophoresis and IHC. However Saykally et al. [21] found a decrease of LAMP2A in the brains of TBI mice models studied with radioimmunoprecipitation 60 days after the trauma.

### 2.7. Cathepsin D (CTSD)

Cathepsin D is a lysosomes endoprotease involved in the degradation of proteins [33]. Sarkar at al. [13] use WB and IF in the brain of TBI rat models to demonstrate that it decreases between 1 and 24 h after trauma and increases between 3 and 7 days after trauma. These alterations were found both in the cortex and hippocampus of mice.

### 2.8. Wnt1 Inducible Signaling Pathway Protein 1 (WISP1)

WISP1 is a protein of the CCN family of the extracellular matrix involved in many signaling pathways comprising apoptosis and autophagy. It also stimulates tumorigenesis [34]. Using PCR for the RNA measure and a combination of WB and IF for the protein measure in the brain of TBI rat models, Ye et al. [17] found that WISP1 and its mRNA decrease 1 day after the trauma in the brain of mice model of TBI, reach the minimum level after 3 days and recover at 5 and 7 days after trauma. It was found in neurons, particularly in SHSY5Y and PC12 cells, of the ipsilateral brain.

### 2.9. β-Catenin

β-catenin is a protein involved in the Wnt pathway. In particular, it interacts with many transcriptional factors to stimulate cell growth and inhibit apoptosis and autophagy [35]. Ye et al. [17] use WB to measure the protein in TBI rat models. They find that the protein expression does not have a linear response to the TBI but decreases between 1 and 7 days after the trauma with a negative peak at 3 days and a partial restoration at 5 days. However in Chen et al. [25] analogue methods in mice models of TBI show an increase of the protein between 6 h and 7 days after the trauma, with a peak after 12 h in prefrontal cortex and an increase between 12 and 24 h in perilesional cortex end hippocampus with a peak at 12 h.

### 2.10. 3-Mercaptopyruvate Sulfurtransferase (3-MST) or tRNA Thiouridine Modification Protein (TUM1)

3-MST is an enzyme involved in cyanide degradation and in thiosulfate biosynthesis expressed in autophagy in response to oxidative stress [36]. Using WB, immunohistochemistry (IHC) and IF in the brain of TBI mice models Zhang et al. [16] found that the protein increases between 6 h and 3 days after trauma with a peak at 1 day. In particular, the protein is found in vital neurons of the ipsilateral cortex.

### 2.11. Transactivation Response Deoxyribonucleic Acid Protein 43 (TDP-43)

TDP-43 is a transcriptional repressor protein that is known to be expressed in autophagy [37,38]. Its research with radioimmunoprecipitation conducted by Saykally et al. [21] show an increased level of protein 12 days after 1 trauma of 50 g and 3 days after 5 traumas of 30 g in ipsilateral cortex. In addition, the protein decreased between 30 and 60 days after 5 traumas of 30 g. However Wisner et al. [22] use IF in the brain of TBI and stab mice models to evaluate alteration of TDP-43 and found an increase of the protein between 3 and 7 days after the trauma with a peak at 3 days after the trauma but only in stab mice models.

### 2.12. Autophagy Related 7 (Atg7)

Atg7 is a protein involved in the conjugation phase of autophagy [38]. Saykally et al. [21] found a reduction of this protein using radioimmunoprecipitation in the brain of TBI mice models WB and IF in the brain of TBI mice models 3 days after the trauma.

### 2.13. Interleukin 33 (IL-33) and Interleukin 1 Receptor-like 1 (IL1RL1) or ST2

IL-33 is a member of the interleukin family and IL1RL1 is an orphan receptor of IL-1 that is known to be an IL-33 receptor too. Both interleukin and receptor are involved in signaling of autophagy [39]. IL-33 was found increased 1 day after a trauma in astrocyte nucleus and oligodendrocyte cytoplasm using WB and IF in the brain of TBI mice models. However IL1RL1 was found to decrease 1 day after a trauma using WB and IF in the brain of TBI mice models [23].

### 2.14. Senescence-Associated-β-Galactosidase (SA-β-gal)

SA-β-gal is a lysosomal enzyme that increases in senescent cells known to be increased after TBI [40]. Using a combination of IHC and IF in the brain of TBI mice models Tominaga et al. [24] found an increase of this protein between 1 and 14 days after the trauma in ipsilateral cortex.

### 2.15. Cyclin D1 and Proliferating Cell Nuclear Antigen (PCNA)

Cyclin D1 is a member of the cyclin protein family and regulates the cell cycle restraining autophagy, while PCNA is a protein involved in DNA replication during cell proliferation [41]. Using a combination of IHC and IF in the brain of TBI mice models, Tominaga et al. [24] found that both proteins increased between 1 and 14 days after the trauma in ipsilateral cortex, with a peak between the fourth and seventh days in particular in neurons, astrocytes, and microglia. Their mRNA, measured with real time-PCR, was also found increased 4 days after trauma.

### 2.16. p16 and p21

p16 and p21 are two inhibitors of cyclin-dependent kinase involved in cell cycle [42,43]. Tominaga et al. [24] studied the variation of these proteins with a combination of IHC and IF in the brain of TBI mice models. They demonstrate that both increase between 4 and 14 days after trauma in ipsilateral cortex. P16 was found in astrocyte and p21 in neurons and microglia.

### 2.17. Chromodomain-Helicase-DNA-Binding Protein 8 (Chd8)

Chd8 is an enzyme involved in chromatin remodeling during cerebral fetal development and its mutations are some of the principal causes of autism. It influences pathways of cell growth and one of its effects is to reduce autophagy [44]. Using IF and WB in the brains of TBI mice models, Chen et al. [25] found a decrease of the molecule 3 h after TBI followed by a partial restoration at 12 h and another decrease until seventh day. However, the enzyme was found increased between 12 and 24 h after TBI in the prefrontal cortex, with a peak at 12 h. A slight increase in the hippocampus was also described between 6 and 24 h, with a peak at 12 h. Remarkably, a double IF localizes the protein in the neurons and not in glial cells.

### 2.18. Sec22b

Sec22b is a sequestrosome membrane protein, member of the SNARE family, which plays an important role in autophagosome trafficking and, consequently, in the autophagy process [45]. Using WB and IF in the brain of TBI rat models Li et al. [26] found a decrease between 12 h and 7 days after trauma with a negative peak at 24 h after trauma in the perilesional cortex.

## 3. Discussion

Autophagy is a stress-response cellular process that allows the controlled degradation of cellular components or the whole cell itself and it has a role in many metabolic processes to maintain the homeostasis of the organism. In fact, it helps to recycle the cellular wastes into resources for the metabolism and remove the damaged or dysfunctional cell organelles even leading to programmed cell death in the case of extreme cell starvation. This process guarantees a progressive degradation of the cell components without harming the surrounding tissues as an alternative to apoptosis. Autophagy can be distinguished in three separate forms: macroautophagy, microautophagy and chaperon mediated autophagy (CMA). In macroautophagy, double-membrane organelles called autophagosome include cell components through various steps and then fuse with lysosomes to enable their degradation; microautophagy and CMA consist in different molecular mechanisms that target cell components and enable their direct phagocytosis in the lysosomes [46,47].

Owing to its involvement in homeostasis maintenance and stress response, autophagy activates in many diseases that affect different organs such as the brain (Alzheimer’s disease, Parkinson’s disease), muscles (myopathies), lungs (cystic fibrosis), heart (cardiac hypertrophy), and it could even be involved in responding to conditions such as infections and obesity [47]. It is also demonstrated that the autophagy process is active in the brain tissue after a TBI, probably as a mechanism of defense from neuronal damage, even if the pathological mechanisms and the physiological functions are not yet totally clear [6,7,8,9,10,11].

Based on this, a literature review has enabled us to define many markers of autophagy that could be used to identify the evidence of a TBI in trauma victims’ brains. In all, 24 markers have been considered and divided into two groups. The first consisted of well-known markers (examined in five or more studies), whose correlation with autophagy is consolidated and used today to test new drugs against negative outcomes of TBI. The second was made of new markers recently proposed in the literature with only a few dedicated studies (involved in fewer than five studies). In particular, the correlation between the presence of the markers from the first group (LC3-II and beclin-1) and trauma in the brain of TBI models seems to be quite certain. These are the first and more widely studied markers considered. Furthermore, these markers are also the most long-lasting ones, so they could be considered the most promising for a first diagnosis of TBI. In fact, they could enable the demonstration of the presence of encephalic trauma even one month after the trauma itself.

In addition to these markers, some of the proteins of the second group, such as connexin and LAMP families, GSK-3β, 3-MST, IL-33, SA-β-gal, cyclin D1 and PCNA, whose increase in the brain of TBI victims is more limited over time, could also be used to study the timing of the trauma. A combination of markers could be particularly accurate to define the elapsed time between trauma and death. However, the correct timing of these proteins has to be studied in the near future because only a few studies exist for each of them and, in some cases, the limits of the elevation of the proteins are caused by the sample timing used by the study and it may not reflect the effective limitations of the protein elevation. The timing of the principal marker for which this study identifies a consensus of elevation after TBI, in TBI models with various techniques, are graphically represented in Figure 1. A summary of the markers and the major finding for each of them is enlisted in Table 2.

However, the use of these markers has to be validated in the future because they are tested almost exclusively in mice and rats, and the only study on the human brain shows poor results [7]. Furthermore, only a few markers are validated in many studies, but for most of them there is only one study enabling the definition of increase-related characteristics. In addition, when more than one study is found, in some cases, contrasting results are presented, for example in the case of p62, β-catenin, TDP-48, and Chd-8.

Moreover, an alteration of the proteins involved in the autophagy process could also be caused by other brain pathological conditions, such as epilepsy [48], ischemic and hemorrhagic stroke [49,50], Alzheimer’s disease [51,52], and Parkinson’s disease [53,54,55,56]. For example, a LC3-II increase and a LAMP2A decrease have been reported in the substantia nigra of Parkinson’s disease models [53], as well as an alteration of GSK-3β and TDP-43 in some forms of dementia [52], and an impairment of LC3-II and p62 in epilepsy [48]. Therefore, it must be evaluated how these diseases, when affecting TBI victims, might influence the result of the markers analysis.

The principal method used to research these protein markers in the studies are WB and IF, so these are the most promising techniques that should be tested in the near future to make a diagnosis of TBI in trauma victims. In addition, IHC, only used in a few cases, deserves to be studied owing to its greater simplicity and low cost.

Finally, it is significant that only the time elapse between trauma and death was evaluated by most studies because samples in mice and rat brains were taken immediately after the sacrifice of the animal. Therefore, the action of post mortal phenomena influence must be considered in future studies. In addition, a sampling map must be drawn up before the human test because different markers are found to be elevated in different cerebral areas.

### Overview of TBI Treatment Targeted on Autophagy Pathways

Another point to concentrate attention on is the treatment strategy focusing on autophagy in TBI. The literature shows the same limitations, but it could be an interesting field of application of markers to study in forensic cases.

The possibility of therapies are various. One is the regulation of circRNAs, useful for improving cognition function, promoting angiogenesis, inhibiting apoptosis, suppressing inflammation, regulating autophagy, and protecting the blood-brain barrier (BBB) in TBI [57]. Therefore, Tetrandrine ameliorated TBI by regulating autophagy to reduce ferroptosis and showed great promise as a potential target in acute CNS injuries [58]. A new therapeutic strategy for TBI is the upregulation of mitophagy to reduce downstream cascades such as cell death, inflammatory response, and oxidative damage [59]. Another kind of therapy could be acupuncture, which has a benign regulatory effect on neuronal autophagy in different stages of TBI, possibly through the mTOR/ULK1 pathway [60]. Ultimately, pharmacological treatment with curcumin [61,62] or dexmedetomidine [63] pointing to reduce inflammation and oxidative stress was also considered as a possible strategy to improve the TBI outcome.

However, these treatments were tested only in laboratory models of TBI in mice and rats, so their efficacy in humans has to be tested in near future to evaluate the possibility of using them in the clinical setting.

## 4. Materials and Methods

A systematic literature review was carried out up to 27 May 2022 on PubMed, following the PRISMA (Preferred Reporting Items for Systematic Reviews and Meta-Analyses) statement’s criteria in accordance with PRISMA 2020 guidelines (Figure 2) [64]. In the identification phase, the terms (“traumatic brain injury” and “autophagy”) or (“traumatic brain injury” and “mitophagy”) were searched in every field without time limits. 241 articles were found and subjected to exclusion criteria that comprehend articles not in English, pre-proof and retracted (12 articles). After that, 229 articles satisfied inclusion criteria. Afterwards, in screening phase, based on the reading of the abstract, only studies concerning protein markers on autophagy in TBI were assessed for eligibility (69 articles). Finally, after a full text review, only the articles concerning autophagy protein markers that demonstrate a correlation with TBI were included in the study.

## 5. Conclusions

In the near future, a combination of histologic autophagy markers could represent a useful method to evaluate the presence of a TBI in the brain of trauma victims and, eventually, to estimate the timing of damage and death. However, some markers and the human application should be validated in future studies.

## Figures and Tables

**Figure 1 ijms-24-00072-f001:**
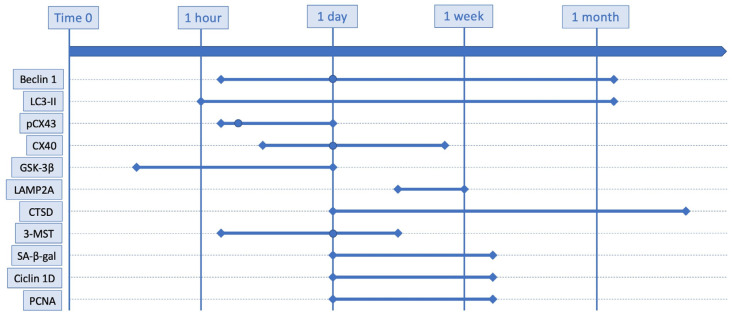
Schematic representation of main markers for which this study identifies a consensus of elevation after TBI, in TBI models with various techniques. Round dots indicate the peak of increase when known.

**Figure 2 ijms-24-00072-f002:**
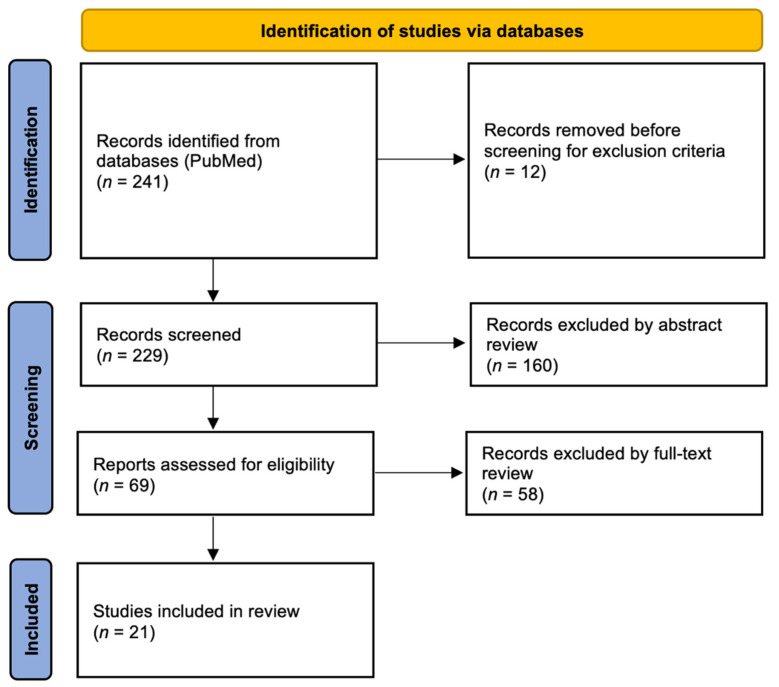
Flow diagram of study design in accordance with PRISMA 2020 guidelines.

**Table 2 ijms-24-00072-t002:** Summary of markers and major findings for each one of them.

Marker	No. of Studies	Conclusion	Models	Techniques
Beclin-1	6	Increased between 4 h and 32 days	Mice, rats, humans	WB, IHC, IF
LC3-II	11	Increased between 1 h and 32 days with a peak at 24 h	Mice, rats, humans	WB, IHC, IF, radioimmunoprecipitation
pCX43	1	Increased between 3 and 24 h with a peak at 6 h	Rats	WB, IF
CX40	1	Increased between 6 h and 6 days with a peak at 24 h	Rats	WB, IF
GSK-3β	1	Increase between 30 min and 24 h	Rats	WB
p62	2	No consensus	Mice	WB, IF
LAMP1	1	Increased between 3 and 7 days	Mice	WB
LAMP2A	3	Increased between 1 and 15 days	Mice	WB, IHC, radioimmunoprecipitation
HSP70	1	Increased between 1 and 15 days	Mice	WB, IHC
CTSD	1	Decreased between 1 and 24 h and increased between 3 and 7 days	Mice	WB, IF
WISP1	1	Decreased between 1 and 7 days with a minimum at 3 days	Rats	WB, IF
β-catenin	2	No consensus	Mice, rats	WB, IF
3-MST	1	Increased between 6 h and 3 days with a peak at 24 h	Mice	WB, IHC, IF
TDP-43	2	Increased between 3 and 7 days	Mice	Radioimmunoprecipitation, IF
Atg7	1	Decreased after 3 days	Mice	Radioimmunoprecipitation
IL-33	1	Increased after 1 day	Mice	WB, IF
ST2L	1	Decreased after 1 day	Mice	WB, IF
SA-β-gal	1	Increased between 1 and 14 days with a peak at 7 days	Mice	IHC, IF
Cyclin-D	1	Increased between 1 and 14 days with a peak between 4 and 7 days	Mice	IHC, IF
PCNA	1	Increased between 1 and 14 days with a peak between 4 and 7 days	Mice	IHC, IF
p16	1	Increased between 4 and 14 days	Mice	IHC, IF
p21	1	Increased between 4 and 14 days	Mice	IHC, IF
Chd8	1	Variable depending on the brain area	Mice	WB, IF
Sec22b	1	Decreased between 12 h and 7 days with a minimum at 24 h	Rats	WB, IF

## Data Availability

All the data used for the review are in the availability of the corresponding author.

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
