# Peer review of "Old and Promising Markers Related to Autophagy in Traumatic Brain Injury"

_ijms, 2022, doi:10.3390/ijms24010072_

Round 1

Reviewer 1 Report

The manuscript is well written, but a few minor changes are required.

1- Please clarify in table one whether LC3 detection in western blots is done as an LC3II/LC3I ratio or only LC3II expression.

2- On page 2 of table one (LAMP1 and LAMP2 increase between 3 and 7 days after trauma in lysosomal and cytosolic fractions), please remove the e and replace it with the & symbol. This is a typo that should be avoided.

3- Please use correct abbreviations, such as western blot-WB, immunofluorescence (IF), and immunohistochemistry (IHC).

4- Could you please explain more about Sec22b?

5-Write one section about treatment strategy focusing on autophagy in TBI and its limitations.

6- In the discussion section, please cite the following article about CMA and autophagy in Parkinson's disease when the authors describe the PD (PMID: 29949106, PMID: 31732923).

Author Response

1- Please clarify in table one whether LC3 detection in western blots is done as an LC3II/LC3I ratio or only LC3II expression.

Thank you for your valuable suggestion. In table one LC3 detection in western blots is done as LC3II expression only. 

2- On page 2 of table one (LAMP1 and LAMP2 increase between 3 and 7 days after trauma in lysosomal and cytosolic fractions), please remove the e and replace it with the & symbol. This is a typo that should be avoided.

Thank you for your kind suggestion. E was replaced with the & symbol.

3- Please use correct abbreviations, such as western blot-WB, immunofluorescence (IF), and immunohistochemistry (IHC).

Thank you for your valuable request. We used the correct abbreviations.

4- Could you please explain more about Sec22b?

Thank you for your valuable request. Some more informations have been added about Sec22b

5-Write one section about treatment strategy focusing on autophagy in TBI and its limitations.

Thank you for your valuable suggestion. A paragraph dedicated to treatment strategy has been added

6- In the discussion section, please cite the following article about CMA and autophagy in Parkinson's disease when the authors describe the PD (PMID: 29949106, PMID: 31732923).

Thank you for your suggestion. Both the references were added.

Reviewer 2 Report

Nil

Author Response

Dear reviewer thank you for spending your time to review the manuscript and share our research.

Reviewer 3 Report

In the manuscript presented by Tommaso Livieri et al., the authors searched the literature and summarized 24 markers related to autophagy after TBI. In this review, each marker was described and discussed one by one. This review is of some significance, but there are a number of issues that need to be addressed.

1. In the Materials and Methods section, the exclusion criteria was only mentioned once but not described. In addition, “160 articles were considered after the application of exclusion criteria” was not correct. The 160 articles were exclude and 69 were considered by abstract review according to Figure 1.

2. In the Results section, two groups of markers were identified. The authors should set criteria for “well-known” markers and “new” markers. There are two Figure 1. Please correct it.

3. There are some grammar errors and non-standard English phrases. The whole manuscript need to be carefully proofread by a native English speaker.

Author Response

1. In the Materials and Methods section, the exclusion criteria was only mentioned once but not described. In addition, “160 articles were considered after the application of exclusion criteria” was not correct. The 160 articles were exclude and 69 were considered by abstract review according to Figure 1. 

Thank you for your valuable request. The exclusion criteria are now described and correct as follow: "241 articles were found and 229 were considered after the application of exclusion criteria. Only articles in English were considered and pre-proof and retracted articles were excluded. After a lecture of title and abstract, 69 articles concerning protein markers on autophagy in TBI were assessed for eligibility. After a full text review, 21 articles were enrolled. Only articles concerning autophagy protein markers that demonstrate a correlation with TBI were included in the study"

In the Results section, two groups of markers were identified. The authors should set criteria for “well-known” markers and “new” markers. There are two Figure 1. Please correct it.

Thank you for your valuable suggestions. We set criteria for well known markers and new markers and we corrected the captions for figures.

There are some grammar errors and non-standard English phrases. The whole manuscript need to be carefully proofread by a native English speaker.

All the manuscripted has been edited from english language mother tongue

Round 2

Reviewer 3 Report

1. The exclusion criteria the authors stated "pre-proof and retracted articles were excluded" is not the true exclusion criteria. Only 21 articles were enrolled. Are all the exclued articles pre-proof and retracted articles?

2. All changed contents in the revised manuscript need to be highlighted in yellow for instance.

Author Response

The exclusion criteria the authors stated "pre-proof and retracted articles were excluded" is not the true exclusion criteria. Only 21 articles were enrolled. Are all the exclued articles pre-proof and retracted articles?

Thank you for your valuable suggestion. Only 12 articles were excluded because retracted or pre-proof. We modified the sentence as follow "A systematic literature review was carried out up to 27th May 2022 on PubMed following the PRISMA (Preferred Reporting Items for Systematic Reviews and Meta-Analyses) statement’s criteria in accordance with PRISMA 2020 guidelines (figure 1)[12]. In the identification phase, the terms (“traumatic brain injury” AND “autophagy”) OR (“traumatic brain injury” AND “mitophagy”) were searched in every field without time limits. 241 articles were found and subjected to exclusion criteria that comprehend articles not in English, pre-proof and retracted (12 articles). After that, 229 articles satisfied inclusion criteria. Afterwards, in screening phase, based on the reading of the abstract, only studies concerning protein markers on autophagy in TBI were assessed for eligibility (69 articles). Finally, after a full text review, only the 21 articles concerning autophagy protein markers that demonstrate a correlation with TBI were included in the study".

All changed contents in the revised manuscript need to be highlighted in yellow for instance

Thank you for your suggestion. We highlighted all changes in yellow.